# DNA Barcoding Supports “Color-Pattern’’-Based Species of *Stictochironomus* from China (Diptera: Chironomidae)

**DOI:** 10.3390/insects15030179

**Published:** 2024-03-06

**Authors:** Chao Song, Guanyu Chen, Le Wang, Teng Lei, Xin Qi

**Affiliations:** 1Zhejiang Provincial Key Laboratory of Plant Evolutionary Ecology and Conservation, School of Life Sciences, Taizhou University, Taizhou 318000, China; songchaonk@163.com (C.S.); 13605875397@163.com (G.C.); leiteng@tzc.edu.cn (T.L.); 2Nanjing Institute of Environmental Sciences, Ministry of Ecology and Environment of China, Nanjing 210042, China

**Keywords:** Chironominae, COI, color pattern, new species

## Abstract

**Simple Summary:**

The Chironomidae family stands out as the most widely dispersed and often the most abundant insect group in freshwater habitats. The significance of color patterns is well-recognized, as they fulfill multiple roles such as communication, camouflage, mimicking, and defense. However, the taxonomy of species within this family, relying on color patterns, as well as the shape and distribution of thoracic pigmentation, wing markings, and leg pigmentation, remains controversial and unstable. Here, we conduct a comprehensive review on the taxonomy of a Chiromidae genus, *Stictochironomus* from China, which is characterized by a combination of distinctive wing and leg markings. Using DNA barcode data and morphological data, two new species to science from China are well supported. Species delimitation analyses performed with distance-based approach and coalescent tree-based approaches also support them as distinct species. Therefore, color patterns should be a good diagnostic characteristic for species delimitation in *Stictochironomus*. Furthermore, we provided an up-to-date taxonomic key for male adults of Stictochironomus from China.

**Abstract:**

The genus *Stictochironomus* (Diptera: Chironomidae) has an almost worldwide distribution, with more than 30 species. However, species delimitation and identification based on the markings on the wings and legs are controversial and uncertain. In this study, we focused on color patterns to review the adults of the genus from China, and two new species (*S. trifuscipes* sp. nov. and *S. quadrimaculatus* sp. nov.) are described and figured. DNA barcodes can accurately separate the two new species with specific color patterns. However, heterospecific individuals form a monophyletic cluster in the phylogeny tree. For example, *S. maculipennis* (Meigen) and *S. pictulus* (Meigen), which have a lower interspecific genetic divergence, form a single clade. Sequences with the same species name but with high intraspecific distance form more than one phylogenetic clade, such as *S. sticticus* (Fabricius) of three clades, *S. pictulus* of four clades, *S. akizukii* (Tokunaga) and *S. juncaii* Qi, Shi, and Wang of two clades, might have potential cryptic species diversity. Species delimitation analysis using ASAP, PTP, and GMYC clearly delineated them as separate species. Consequently, color patterns are a good diagnostic characteristic for species delimitation in *Stictochironomus*. The distance-based analysis shows that a threshold of 4.5–7.7% is appropriate for species delimitation in *Stictochironomus*. Additionally, an updated key including color pattern variation for male adults of known *Stictochironomus* species from China is provided.

## 1. Introduction

Color patterns play a significant role in the behavior of insects and are recognized as significant characteristics for taxonomic studies [1]. Among insects, for instance, the wing color pattern has been important in the evolution of beetles, flies, and moths and butterflies, evolving various functions such as attracting mates and warning predators. One well-studied pattern is butterfly wings, which focuses on the scale-covered wings, pigmentation, and spatial pattering systems [2]. Phenotypic variation significantly conceals the cryptic species diversity that is largely hidden within species based on morphology alone. Theoretically, pigmentation intensity in the body, legs, and wings varies widely, often distinguishing sexes, populations, and species [3].

DNA barcoding provides new insights for exploring species diversity and delimitation, and extensive sampling for meta-barcodes has revealed high cryptic diversity [4,5,6,7,8,9,10]. In 2004, Hebert et al. discovered ten cryptic species in a neotropical skipper butterfly named *Astraptes fulgerator* (Walch) using DNA barcoding and morphological traits, including caterpilal color patterns and subtle differences in adult patterns and size [7]. In 2017, Janzen et al. employed complete nuclear genome sequencing to obtain DNA barcoding data, the ecological distribution, the natural history, and the subtleties of adult color pattern and size, to reveal the cryptic diversity among the species of *Udranomia kikkawai* (Weeks) [8].

Chironomidae have evolved a diverse range of pigmentation patterns on different body parts, including the abdomen, thorax, wings, and legs [11,12,13]. The genus *Stictochironomus* is widely distributed, occurring in all biogeographical regions except Antarctica, and is characterized by a combination of distinctive wing and leg markings, lack of frontal setae, conical scutal tubercle, fused tibial combs, usually with only one spur, and a distally flattened gonostylus with mobile inferior volsella bent laterally [13]. The larvae of this genus can be found in profundal soft sediments or littoral sand of oligotrophic lakes, as well as in sediments of streams and slowly flowing rivers [14]. There are more than thirty species worldwide, with five species recorded in China [15,16,17,18].

In this study, we review the genus *Stictochironomus* and describe two new species from China. Illustrations based on adult males, diagnoses and descriptions based on morphology, and DNA barcodes are also included. A key to the known adult males of this genus is provided. 

## 2. Materials and Methods

The examined material was caught using light traps and then preserved in 75% ethanol at 4 °C in a refrigerator before morphological and molecular procedures. Genomic DNA was extracted from the thorax and head following Frohlich et al. (1999) [19] at Taizhou University, Zhejiang, China. The COI sequences were amplified using the universal primers [20]. For the PCR, 12.5 µL of 2 × Es Taq MasterMix (CoWin Biotech Co., Beijing, China), 0.625 µL of each primer, 2 µL of template DNA, and 9.25 µL of deionized H_2_O were used. PCR amplification was performed with the following program: 95 °C for the initial denaturation for 5 min; 34 cycles of 94 °C for 30 s, 51 °C for 30 s, and 72 °C for 1 min; and a final extension for 3 min at 72 °C [21]. After the PCRs, 2.5 µL of PCR products were run on a 1.0% agarose gel, purified using ExoProStar 1-Step, and sequenced at the Beijing Genomics Institute Co., Ltd., Hangzhou, China.

After genomic extraction, the exoskeletons were slide-mounted in Euparal together with corresponding body parts following the procedure described by Sæther (1969) [22].

The type materials, including holotype and paratypes of the newly described species, were deposited in the collection of the College of Life Sciences, Taizhou University, Zhejiang, China (TZU). The morphological terminology employed in this study follows that of Sæther (1980) [23]. Images of the habitus of each specimen were captured using a DV500 5MP Digital Camera attached to a stereo microscope (Chongqing Optec Instrument Co., Ltd., Chongqing, China). Images of the mounted specimens were captured using a compound microscope from Leica DMLS (Leica Camera AG, Wetzlar, Germany). Photograph post-processing was carried out in Adobe Photoshop and Illustrator version 8 (Adobe Inc., San Jose, CA, USA).

In addition to our own data (GenBank accession numbers: OR822593-OR822597), the available *Stictochironomus* COI barcodes in BOLD were searched, and 704 sequences were added to the dataset named “Barcodes of *Stictochironomus* species (DS-STICTO)”, DOI: dx.doi.org/10.5883/DS-STICTO in the Barcode of Life Data System (http://www.boldsystems.org/ (accessed on 12 December 2023)) [24]. Three datasets were compiled for the analysis: dataset (A) raw data (704 sequences, Appendix A); dataset (B) without duplicated sequence (218 sequences, Appendix A); and dataset (C) holotype sequences (75 sequences, Appendix A). Datasets A and C were only used for ASAP analysis.

Raw sequences were assembled in BioEdit 7.2.5 [25]. The sequences were aligned using the ClustalW algorithm in MEGA 7. The pairwise distances were calculated using the Kimura 2-Parameter (K2P) substitution model in MEGA 7. The neighbor-joining tree was constructed using the K2P substitution model, and 1000 bootstrap replicates and the “pairwise deletion” option for missing data were utilized [26]. Partitions and model selection were performed in PartitonFinder using greedy search and selected according to the BIC scheme [27]. The results were as follows: SYM + G for the first site, F81 + I for the second site, and GTR + I + G for the third site.

For ML analysis, RAxML allows for only a single model of rate heterogeneity in partitioned analyses; we used the GTR + G + I substitution model with 1000 bootstrap replicates in a rapid bootstrap analysis [28]. Bayesian inference analysis was carried out using Markov Chain Monte Carlo (MCMC) randomization in MrBayes v3.2.7 [29] with five million generations, and it was stopped when the standard deviation of the split frequencies was below 0.01. A consensus tree was summarized with the first 25% of sampled trees being discarded as burn-in. Trace files of the BI analysis were also inspected in Tracer 1.7 [30], and then the tree was visualized in FigTree v.1.4.2.

ASAP (assemble species by automatic partitioning) analysis was performed on the website https://bioinfo.mnhn.fr/abi/public/asap/asapweb.html (accessed on 19 April 2023) using the K2P model with other parameters the default setting [31]. The bPTP analyses were run on online servers (http://species.h-its.org/ptp, accessed on 24 January 2024) and the input ML tree generated in RAxML. We ran 500,000 MCMC generations, used a burn-in of 0.1, and set the other parameters to their defaults [32]. The general mixed Yule-coalescent (GMYC) method was applied using the splits package [33] in R. The input ultrametric tree for GMYC was generated in BEAST 1.7 [34] (strict clock, MCMC chain using 10 million generations, TN93 substitution model, and Yule speciation model). Ther convergence and estimated effective sample sizes (ESS > 200) for all the parameters were checked using Tracer v1.7.1 [30]

## 3. Results and Discussion

### 3.1. Barcode Analysis

The 218 aligned sequences ranged from 480 to 658 pairs, among which 252 were variable sites (223 sites parsimony informative; Table 1). Most of the variable sites occurred in the third codon position.

The average intraspecific divergence was 4.27%, which is higher than that in previous studies (0.9–2.8) [12,35,36,37,38,39]. The high intraspecific divergence in our study can be attributed to four species (*Stictochironomus sticticus* (Fabricius): 16.81%; *Stictochironomus pictulus* (Meigen): 14.41%; *Stictochironomus akizuki* (Tokunaga): 15.56%; and *Stictochironomus juncaii* Qin, Shi and Wang: 9.35%; Table 2). Some of these highly divergent species appeared paraphyletic or polyphyletic in the phylogeny tree (Figure 1).

The interspecific divergences ranged from 1.11% to 19.25%, with the maximum between *S. akizukii*-Clade 2 and *S. devinctus* and minimum between *S. maculipennis* and *S. pictulu*-clade 3 (Appendix A). Incorrect identification can occur when two distinct species have an interspecific genetic divergence of 2–3%. As the vouchers cannot be checked, we compared their pigmentation, but the vouchers (NHRS-BYWS00000113, and NHRS-BYWS00000114) were not uploaded to BOLD. These vouchers should be rechecked to determine whether incorrect identifications are responsible for the high intraspecific divergence that we observed for some *Stictochironomus* species.

*Stictochironomus* species exhibit great phenotypic plasticity, especially in terms of the color patterns (leg, wing, and thorax) and the numbers of setae on the thorax (dorsocentrals, prealars, and scutellars), while the permutation and combination of markings or patterns are sometimes easily overlooked by taxonomists who are more concerned with the structural characteristics of the hypopygium. In our study, we did not observe interspecific overlap of color pattern morphotypes, which could support the idea that color patterns are a good diagnostic characteristic for species delimitation in *Stictochironomus*. However, species with unique color morphotypes could form several separated clades based on genetic analyses. Four species with unique color patterns formed several phylogenetic clades (Figure 1), which matches the results in the thesis of Reistad (2023) [40]. These four species were the same species with high intraspecific divergence and included *S. pictulus* (four clades), *S. sticticus* (three clades), *S. akizukii* (two clades), and *S. juncaii* (two clades). There are also cases of molecular discordance in which morphospecies have commonly been found in insect groups because of incomplete lineage sorting, introgression, and hybridization, but these rarely pose problems in relation to chironomids [41]. Therefore, inaccurate reference taxonomy mainly explains the non-monophyletic phenomena of Chironomidae. Reistad (2023) argued that one of the most notable differences was the number of setae on the thorax for differentiating *S. pictulus* clusters as well in some other species, which would support undescribed taxa.

### 3.2. Phylogenetics and Species Delimitation

The neighbor-joining (Appendix A), maximum likelihood (Appendix A), and Bayesian inference methods of constructing phylogenetic tress almost share the same topology (Appendix A). The high interspecific variation in COI means that the relationships between species are not well constructed when it is used [42]. However, the large interspecific variation aids in the diagnosis of species clusters, resulting in high support values at the ends of the branches. Based on the NJ tree, ML tree, and BI tree, 218 DNA barcodes of 16 initial morphospecies were clustered into 22 clades (Figure 1). The undescribed taxa well form separated clades in all the phylogenic trees, which supports them as new to science. 

Species delimitation analyses performed with the distance-based approach and coalescent tree-based approaches almost produced the same results. In the Assemble Species by Automatic Partitioning analysis, dataset A gave the lowest score of 2.5, with the best partition of 26 species (threshold at 4.9%, Appendix A); dataset B gave a score of 4.0, with the best partition of 21 species (threshold at 6.9%, Appendix A); and dataset C gave a score of 3.0, with the best partition of 20 or 24 species (threshold at 7.7% or 4.5%, Appendix A). Using the Poisson Tree Processes (PTP) model to infer putative species, under the maximum likelihood solution, the highest Bayesian-supported solution returned 26 species, while the confidence interval was 26–97 (mean: 40) OTUs (Appendix A). The single-threshold general mixed Yule-coalescent calculations (ST-GMYC) yielded 30 entities, ranging from 30 to 30 (Appendix A). In summary, in the results for ASAP, GMYC, PTP, and phylogeny clades, the species numbers varied from 20 to 30, more than the pre-morphospecies number. The two undescribed species (*Stictochironomus trifuscipes* and *S. trifuscipes*) are always regarded as separated species, which supports them as new to science.

Most barcoding research on insect species proposes a threshold of 2–3% for delimiting intra- or interspecific species [42,43,44]; however, genus-specific thresholds have been suggested for some Chironomidae including 4–5% for *Tanytarsus* [36] and 5–8% for *Polypedilum* [21]. In this study, a barcode gap was explored in *Stictochironomus* species using three datasets (dataset A: sequences with duplicates; dataset B: sequences without duplicates; dataset C: haplotype sequences) (Figure 2). Barcode gaps appeared at 1–7%, 2–7%, and 3–4%, respectively, which marks the limit between smaller intraspecific distances and larger interspecific distances. From the gap, a distance threshold is estimated and used to partition the samples into putative species. The threshold values generated from ASAP ranged from 4.5 to 7.7%, with species numbers ranging from 20 to 26, which approximately align with the phylogenetic clusters. Therefore, a threshold of 4.5–7.7% is more appropriate for species delimitation in *Stictochironomus* species.

Consequently, for future research, more genetic markers are needed to test whether the phylogenetic units are congruent with biological units. On the other hand, the morphological differences between closely related species are often subtle, and information from more than one life stage and data on behavior and ecology tend to be very helpful for species delimitation.

### 3.3. Taxonomy

#### 3.3.1. *Stictochironomus quadrimaculatus* Song and Qi, sp. nov.

urn:lsid:zoobank.org:act:EC674AA4-48E7-4D2E-A869-B00FE884EA98

(Figure 3, Figure 4 and Figure 5A; GenBank accession: OR822595)

Type material. Holotype: male (BOLD and TZU sample ID: ZJCH179; Field ID: BSZ46), China, Zhejiang Province, Lishui City, Qingyuan County, Baishanzu National Nature Reserve, N 27.581944, E 117.154722, 12 August 2020, leg. C. Song, light trap.

Diagnostic characteristics. The male imago can be separated from the known species of *Stictochironomus* based on the following combination of features: antenna and antennal plumage mostly yellowish brown; wing with six separate markings, two spots in Cell m_1+2_; thorax yellowish brown or brownish; legs pigmented, especially four dark rings on all tibia; superior volsella sickle-shaped, with three basal long inner setae and microtrichiae at the base; and a distal seta arising at the apical 1/3.

Etymology. The new species is named based on the characteristics of the color patterns of tibia. In Latin, it consists of “*quadri*” and “*maculatus*”. The Latin word “*quadri*” means “four”, while “*maculatus*” means “spotted” or “banded”. This term is used to describe the features of species that have four dark bands on the tibia.

Description. Male imago (*n* = 1). Total length 3.75 mm. Wing length 1.80 mm. Total length/wing length 2.08. Wing length/length of profemur 1.71.

Coloration (Figure 3). A mature male adult is mostly pale yellowish to light brown; the antenna is yellowish brown and antenna plumage yellowish; the ground of th thorax is yellowish brown with dark lateral stripes, anepisternum II, and the postnotum is dark brown; the abdomen is yellowish; the wings are without markings; the legs have poorly defined pigmentation. P1: mostly pale, the anterior part of the femur has a pale brown ring, the tibia is slightly pale brown, tarsus pale. P2 and P3: pale, except the knees are slightly brown. 

Head. AR 1.44. The head was destroyed for genome exception except for the antenna.

Thorax (destroyed). Acrostichals more than 3; dorsocentrals 7; prealars 6; scutellars 5. Scutral tubercle present.

Wing (Figure 4A). VR 1.01. Distribution of setae on veins: R, 19; R_1_, 18; R_4+5_, 35. Squama with 13 setae. The anal lobe is normally developed.

Legs (Figure 5A). The spur on the median tibiae is 27 µm long, including an 18 µm long comb and the un-spurred comb is 18 µm long; the spur on the hind tibia is 32 µm long including a 20 µm long comb and the un–spurred comb is 15 µm long. The width at the apex of the front tibia is 53 µm; the width of the middle tibia is 53 µm; and the width of the hind tibia is 57 µm. The lengths (in µm) and proportions of the legs can be found in Table 3.

Hypopygium (Figure 4B,C). The anal point is straight and parallel in the dorsal view, 88 µm long and 37 µm wide at the base and 6 µm wide at the apex. Tergite IX with 9 setae medially; Laterosternite IX with 2–3 setae. The transverse sternapodeme is 55 µm long and the phallapodeme is 77 µm long. The gonocoxite is 192 µm long and the gonostylus is 130 µm long. The superior volsella is sickle-shaped, 77 µm long, 30 µm wide, with a 3-basal long inner seta and microtrichiae in the base and 1-distal seta arising at apical 1/3. Inferior volsella with 12 setae, is 100 µm long, and extends beyond the apex of anal point. HR 1.48; HV 2.88.

Distribution. China (Zhejiang).

Remarks. The new species is similar to *Stictochironomus han* Na and Bae, 2010. Both species share the following characteristics: wing with obvious separated spots; an anal point that is long, slender, nearly parallel-sided, and with a rounded apex; a superior volsella that is broad basally with microtrichiae and three long setae on the inner margin and with the long distal process slightly curved, bare, with a hooked apex, and a long seta on the dorsal surface in distal 1/3 [45]. *Stictochironomus quadrimaculatus* sp. nov. differs from *S. han* by a lower AR (1.44), shorter wing (1.80 mm), larger foreleg LR (1.88), and having unique pigmentation patterns on the legs and wings, such as spot locations, sizes, and numbers (Figure 1A,D in [43]). In *S. han*, these measurements are AR 2.48, the wing length is 3.04 mm, and the foreleg LR is 1.12 (Table 4). Moreover, the average species k2p distance to the other *Stictochironomus* ranges from 15% to 22% (Appendix A), which to some extent, supports it as a new species.

#### 3.3.2. *Stictochironomus trifuscipes* Song and Qi, sp. nov.

urn:lsid:zoobank.org:pub:7B3EC4A2-83A9-4618-91DA-75C8022EAB96

(Figure 5B, Figure 6 and Figure 7; GenBank accession: OR822594)

Type material. Holotype: male (BOLD and TZU sample ID: ZJCH206; Field ID: BSZ73), China, Zhejiang Province, Lishui City, Qingyuan County, Baishanzu National Nature Reserve, N 27.581944, E 117.154722, 12 August 2020, leg. C. Song, light trap; Paratype: 1 male, China, Fujian Province, Nanping City, Wuyishan County, Wuyi Mountain National Forest Park, N 27.601667, E 117.789167, 16 April 2021, leg. K.H. Zhong, light trap.

Diagnostic characteristics. The male imago can be separated from the known species of *Stictochironomus* based on following combination of features: the antenna and antennal plumage is mostly dark brown; the wing is smokey with markings; the thorax is yellowish brown or brownish with the postnotum dark brown; the legs are pigmented, especially the fore Ta_1–3_ which is dark brown; the superior volsella sickle-shaped, with three basal long inner setae and microtrichiae at the base and one distal seta arising at the apical 1/3.

Etymology. The new species is named based on the characteristics of the color patterns of the fore tarsi. In Latin, it consists of “*tri*” and “*fuscipes*”. The Latin word “*tri*” means “three”, while “*fuscipes*” means “dark legs”. This term is used to describe the feature of species where the first three segments of the fore tarsi are dark in color.

Description. Male imago (*n* = 2). Total length 4.88–5.28 mm. Wing length 2.25–2.33 mm. Total length/wing length 2.17–2.27. Wing length/length of profemur 1.73–1.88.

Coloration (Figure 6 and Figure 7). A mature male adult is mostly brownish or yellowish brown. The head is brownish or dark brown; the antenna and antenna plumage are dark brown; the thorax is yellowish brown or brownish with the postnotum dark brown. The Abdomen is yellowish or brownish. Legs. P1: the femur is not clearly defined and is mostly pale with two brown rings; most of the tibia is pale with the anterior 2/5 brownish; ta_1–3_ is dark brown and distal 2/5 of ta_4_ and distal 1/2 of ta_5_ are brown. P2: most of the femur is brownish with three pale rings, most of the tibia is dark brown with two pale parts, and most of the tarsus is pale, with basal 1/4 and apical 1/5 of ta_1_ and apical 1/2 of ta_4_, and ta_5_ being dark brown. P3: most of the femur is dark brown with three pale rings, the tibia is dark brown except for a pale basal part; the tarsus is pale except for ta1, apical 1/2 of ta4, and ta5, which are dark brown.

Head. AR 1.54 (*n* = 1). The temporal setae are 10–13, including 5–6 inner verticals, 4–6 outer verticals, and 1 postorbital. The clypeus is 19–22 setae. The tentorium is 150–167 μm long and 43–45μm wide. The palpomere length is as follows (in μm): 50–75, 88–88, 175–187, 160–63, 287(*n* = 1). The palpomere ratio (5th/3rd) is 1.64.

Thorax. Acrostichals 8–12; dorsocentrals 17–20; prealars 4–5; Scutellum has 10–10. Scutellar tubercles present.

Wing (Figure 7A). VR 1.05–1.06. The distribution of setae on the veins is as follows: R, 20–21; R_1_, 11–20; and R_4+5_, 22–31. The squama has 11–12 setae. The anal lobe is normally developed. 

Legs (Figure 5B). The spurs on the median tibiae are 25–30 µm long, including a 25–32 µm long comb and an un-spurred comb that is 32–35 µm long; the spur on the hind tibia is 30–32 µm long including a 30–32 µm long comb and an un–spurred comb that is 32–35 µm long. The width at the apex of the front tibia is 73–78 µm; the middle tibia is 65–78 µm; and the hind tibia is 63–65 µm. Lengths (in µm) and proportions of legs are shown in Table 5.

Hypopygium (Figure 7B,C). The anal point is straight and parallel in the dorsal view, 95–115 µm long and 27–27 µm wide at the base and 5–5 µm wide at the apex. Tergite IX has 6–11 setae medially and laterosternite IX has 2–3 setae. The transverse sternapodeme is 55–60 µm long and the phallapodeme is 75–75 µm long. The gonocoxite is 260–275 µm long and the gonostylus is 135–140 µm long. The superior volsella is sickle-shaped, 65–75 µm long, 28–30 µm wide, with three basal long inner setae and microtrichiae at the base and a distal seta arising at the apical 1/3. The inferior volsella has 18–20 setae and is 110–120 µm long, not extending beyond the apex of the anal point. HR 1.92–1.94; HV 3.61–3.71.

Distribution. China (Fujian, Zhejiang).

Remarks. The new species is similar to *Stictochironomus simantomaculatus* (Sasa, Suzuki and Sakai, 1998). Both species have similar wing patterns and pigmentation of the fore femur and fore tibia, but they can be differentiated by the morphology of the hypopygium. Specifically, the superior volsella has three inner setae and one lateral outer seta in *S. trifuscipes* sp. nov. versus two inner setae and zero lateral outer setae in *S. simantomaculatus*. In addition, the gonostylus is normally developed in *S. trifuscipes* sp. nov. and abruptly constricted at about the middle in *S. simantomaculatus*. The wing patterns and superior volsellae of *Stictochironomus trifuscipes* sp. nov. also resemble *Stictochironomus multannulatus* (Tokunaga, 1938) (Figures 38 and 39 in [46]), but these species can be distinguished by leg pigmentation and details on the inferior volsella. For example, ta1–3 of P1 is totally dark brown (Figure 5B), while it is mostly pale in the latter species. In addition, the inferior volsella has 18–20 recurved setae versus 16 setae; the gonostylus lacks long strong setae along the inner margin but has strong setae on the ventral surface and several long bristles arising from the basal half (Table 6). In addition, the average species k2p distance to the other *Stictochironomus* ranged from 16% to 24% (Appendix A), which to some extent, supports it as a new species.

#### 3.3.3. *Stictochironomus akizukii* (Tokunaga, 1940)

*Chironomus (Stictochironomus) akizukii* Tokunaga,1940:299 [47].

*Stictochironomus akizukii*, Sasa 1985:38[48]; Wang 2000: 647[49].

Specimens examined. One male, China, Tianjin City, Ji County, Yuqiao Lake, 30 March 1988, leg. X.H. Wang, light trap; one male, China, Inner Mongolia Autonomous Region, Moerdaoga County, 8 July 1988, leg. W.J. Bu, light trap; three males, China, Sichuan Province, Ganzi Prefecture, Daocheng County, Daocheng River, 11 July 1996, leg. Y.Z. Du, light trap; five males, China, Yunnan Province, Lijiang City, Baishuihe River, leg. Y.Z. Du, light trap.

Diagnosis. The male imago can be separated from the known species of *Stictochironomus* based on the following combination of features: wing with only one dark area around the cross vein r-m; superior volsella with four basal setae; gonostylus with twelve long setae along the inner margin; a thorax yellow with scutum, postnotum, and scutellum brown; and the abdominal tergites I–IV pale yellow, with apical brown band, tergites V–IX brown.

Distribution. China (Tianjin, Inner Mongolia, Sichuan, Yunan), Japan.

#### 3.3.4. *Stictochironomus juncaii* Qi, Shi, and Wang, 2008

*Stictochironomus juncaii* Qi, Shi, and Wang, 2008: 281 [18].

Specimens examined: two males, (TZU: CH195, CH214), China, Jilin Province, Yianbian City, Fusong County, Nenjiang Park, 42.17388° N 127.515° E, 12.VII.2016, leg. C. Song; one male (TZU: CH212), China, Heilongjiang Province, Yichun City, Tangwang County, N 47.73083, E 128.8953, 12 July 2016, leg. C. Song.

Diagnostic characters. The male imago can be separated from the known species of *Stictochironomus* based on following combination of features: wing with several dark spots, posterior area somewhat smoky; the anal point slender and parallel-sided; the gonostylus robust, basal gonostylus sturdy, gradually narrowed to apical, with 13 strong setae along the inner margin.

Distribution. China (Heilongjiang, Jilin, Liaoning), Palearctic region.

#### 3.3.5. *Stictochironomus multannulatus* (Tokunaga, 1938)

*Chironomus* (*Polypedilum*) *multannulatus* Tokunaga, 1938: 339 [50].

*Stictochironomus multannulatus* (Tokunaga), Sasa 1984: 51 [46]; Sasa 1989: 28 [51]; Yamamoto 1980: 24 [52].

Specimens examined. One male, China, Jiangxi Province, Poyang Lake, 12 June 2004, leg. C.C. Yan, light trap; one male, China, Guizhou Province, Maolan Mountain, 29 July 1995, leg. W.J. Bu, light trap.

Diagnosis. The male imago can be separated from the known species of *Stictochironomus* by the following combination of features: wing smoky, with transverse markings; legs with conspicuous dark and pale rings, as the tibia is mostly black with a preapical pale ring, ta_1–3_ mostly pale with apical dark; superior volsella hooked, with 3–4 setae at the base and one dorsal seta arising at the apical 1/3.

Distribution. China (Jiangxi, Guizhou), Japan, Russian Far East [53].

#### 3.3.6. *Stictochironomus pictulus* (Meigen, 1830)

*Chironomus pictulus* Meigen, 1830: 244 [54].

*Stictochironomus pictulus* (Meigen), Pinder 1978: 140 [55]; Yamamoto 1980: 24 [50]; Sasa and Suzuki 1998: 22 [56].

Specimens examined. One male, China, Xinjiang Uygur Autonomous Region, Ili Kazak Autonomous Prefecture, Tekes County, 4 August 2002, leg. H.Q. Tang, light trap; two males, China, Xinjiang Uygur Autonomous Region, Kashi City, 17 August 2002, leg. H.Q. Tang, light trap.

Diagnosis. The male imago can be separated from the known species of *Stictochironomus* by the following combination of features: wing with four separated spots; abdominal tergites mostly black but each tergite with a pale band along posterior margin; legs with conspicuous dark and pale rings, as the femora are largely dark and have a short preapical pale ring; tibiae with three dark rings and two pale rings; superior volsella with a narrow base bearing four inner setae; and a narrow, apically hooked and pointed distal horn bearing a long lateral seta arising at near the tip.

Distribution. China (Xinjiang); Holarctic region.

#### 3.3.7. *Stictochironomus sticticus* (Fabricius, 1781)

*Tipula sticticus* Fabricius*,* 1781: 407 [57].

*Stictochironomus histrio* Sasa, 1985: 115 [48].

*Stictochironomus sticticus* (Fabricius, 1781) Pinder, 1978: 140 [55].

Specimens examined. Three males, China, Hebei Province, Weichang County, 14 June 2001, light trap, Y.H. Guo; sixteen males, China, Qinghai Province, Menyuan County, 16 June 1989, X.H. Wang, light trap.

Diagnosis. The male imago can be separated from the known species of *Stictochironomus* based on following combination of features: wing with only one dark area around the cross vein r-m; legs with conspicuous dark and pale rings, as the fore femur is yellowish, with two apical dark brown rings, the mid- and hind femur brown, with one apical pale ring; superior volsella with a narrow base bearing 6–7 inner setae; and a narrow, apically hooked and pointed distal horn bearing a long lateral seta arising at distal 1/3.

Distribution. China (Hebei, Qinghai); Holarctic region.

An updated key for the known males of *Stictochironomus* from China.

The following key is modified from Qi et al. (2008) [18], with color pattern variations seen in Appendix A.

1.Wing with only one dark area around cross vein r-m··········································································································································································································································2-Wing membrane with several dark spots································································································································································································································································32.Superior volsella with six or more basal setae; gonostylus slender and cylindraceous, with nine short setae concentrating in the apex; thorax and abdomen brown (Figures 8 and 9 in [48])··································································································································································································*S. sticticus* (Fabricius)-Superior volsella with four basal setae; gonostylus with twelve long setae along inner margin; thorax yellow with scutum, postnotum and scutellum brown; the abdominal tergites I–IV pale yellow, with apical brown band, tergites V–IX brown (Figure 35 in [46])························································································································································································*S. akizukii* (Tokunaga)3.Wing with smoky spots ··························································································································································································································································································4-Wing with more than four clearly separated spots················································································································································································································································54.Inferior volsella with 14 setae; dorsocentrals 10–15, acrostichals 14–20, scutellars 28; fore tabia mostly black, with 1 subapical pale ring (Figure 39 in [46]) ···············································································*S. multannulatus* (Tokunaga)-Inferior volsella with 18–20 setae; dorsocentrals 17–20, acrostichals 8–12, scutellars 10; fore tibia mostly pale, with anterior 2/5 brownish (Figure 5B)······································*S. trifuscipes* Song and Qi sp. nov.5.Posterior area of wing without smoky spots························································································································································································································································6-Posterior area of wing somewhat smoky, with five spots; dorsocentrals 18, acrostichals 12, scutellars 25; inferior volsella with 20 setae; tibiae with three dark rings and two pale rings (Figure 1 in [18]) ··································································································································································································*S. juncaii* Qi, Shi, and Wang6.Wing with six spots; dorsocentrals 7, scutellars 5; inferior volsella with 12 setae; fore femur largely pale with three dark ring, tibia with four dark rings (Figure 5A) ····················································································································································································································*S. quadrimaculatus* Song and Qi sp. nov.-Wing with four spots; dorsocentrals 18–22, acrostichals 25–18, scutellars 28–32; inferior volsella with 20–23 setae; fore femur largely brown with a short preapical ring, tibia with three dark rings (Figure A15 in [56]) ··································································································································································································*S. pictulus* (Meigen)

## 4. Conclusions

Our study supports the idea that color patterns are an important diagnostic characteristic for species delimitation in Chironomidae species. By combing morphology and DNA barcode data, two species new to science were discovered and described after reviewing the *Stictochironomus* species from China. However, a phylogeny tree based on public barcode data indicated that conspecific individuals formed more than one clade (e.g., *S. sticticus* of three clades and *S. pictulus* of four clades), which might indicate potential cryptic species diversity. Species delimitation analyses using ASAP, PTP, and GMYC well support the two new undescribed species to science. The ASAP results indicate that a threshold of 4.5–7.7% is more appropriate for species delimitation in *Stictochironomus*. As it may not be possible to check the vouchers registered in BOLD or GenBank, we urge scientists to confirm their identifications with experts and to upload Appendix A, such as closeup images of voucher specimens and habitat descriptions.

## Figures and Tables

**Figure 1 insects-15-00179-f001:**
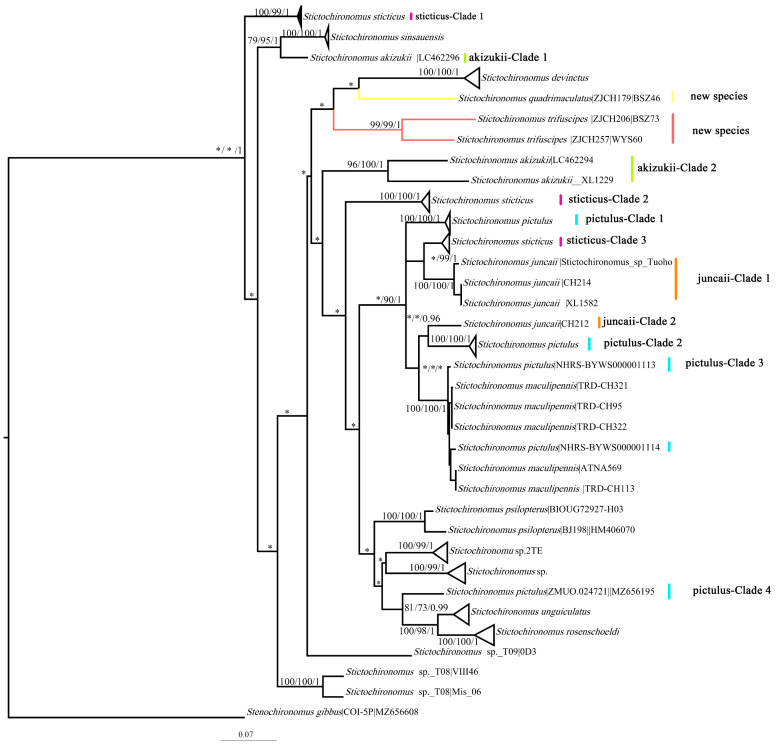
Phylogeny tree for *Stictochironomus* based on DNA barcode sequences. The numbers on the branches refer to the bootstrap support, varying from 75 to 100/ML; bootstrap values vary from 70 to 100 and posterior probabilities vary from 0.95 to 1 (* means that no support indicated on nodes); different colors mean different species.

**Figure 2 insects-15-00179-f002:**
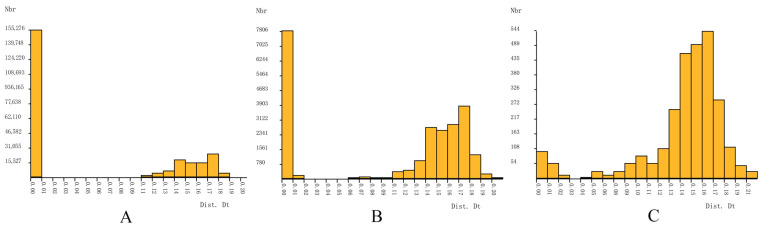
Histograms of *Stictochironomus* sequences generated by ASAP. (**A**) Raw data (704 sequences); (**B**) dataset without duplicated sequence (218 sequences); (**C**) holotype sequences (75 sequences).

**Figure 3 insects-15-00179-f003:**
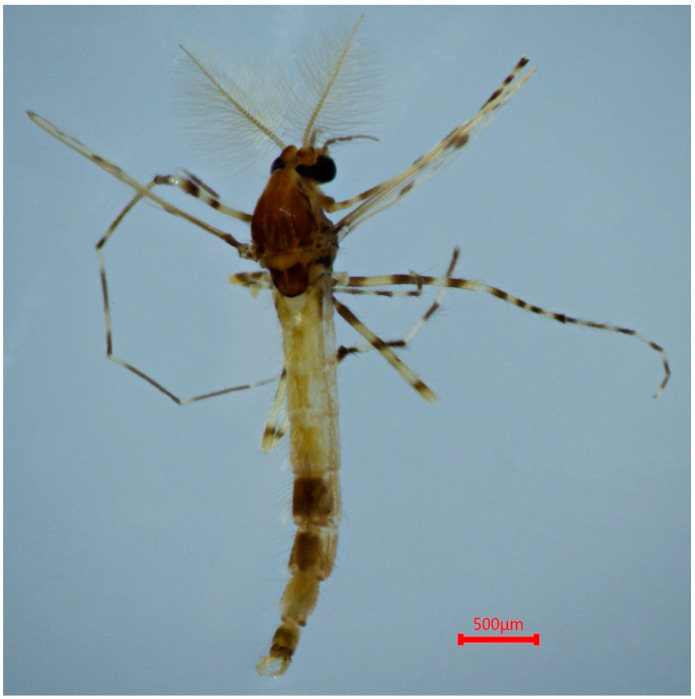
Male adult (holotype, in dorsal view) of *Stictochironomus quadrimaculatus* sp. nov.

**Figure 4 insects-15-00179-f004:**
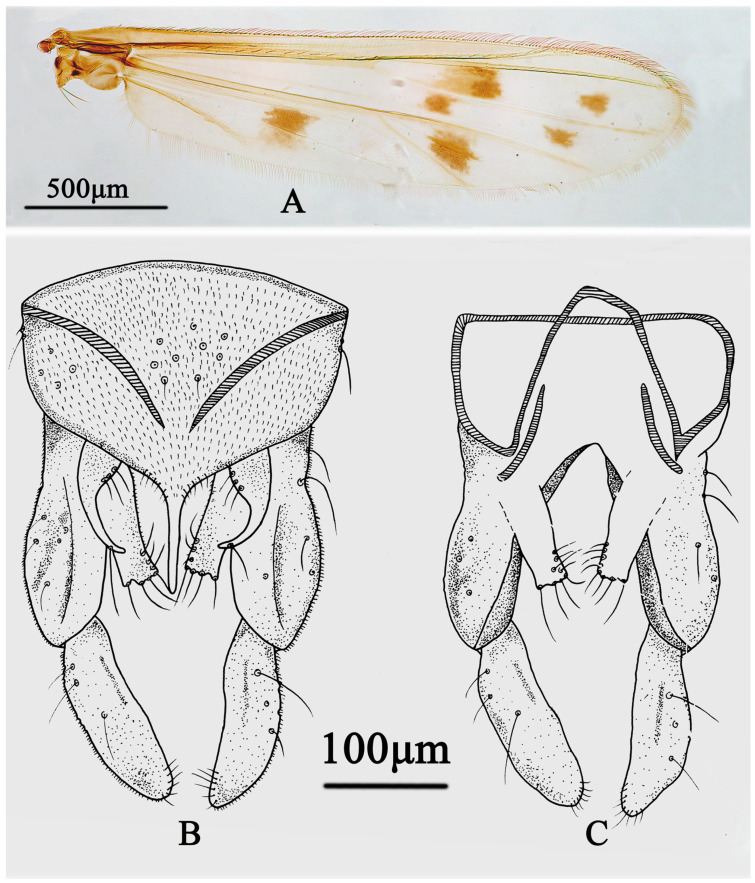
Male adult of *Stictochironomus quadrimaculatus* sp. nov.: (**A**) wing; (**B**) hypopygium in dorsal view; (**C**) hypopygium in ventral view.

**Figure 5 insects-15-00179-f005:**
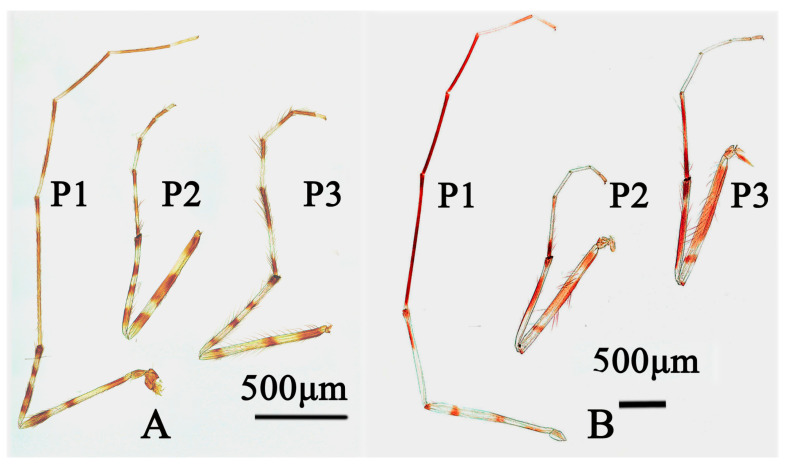
legs. (**A**) *Stictochironomus quadrimaculatus* sp. nov.; (**B**) *Stictochironomus trifuscipes* sp. nov.

**Figure 6 insects-15-00179-f006:**
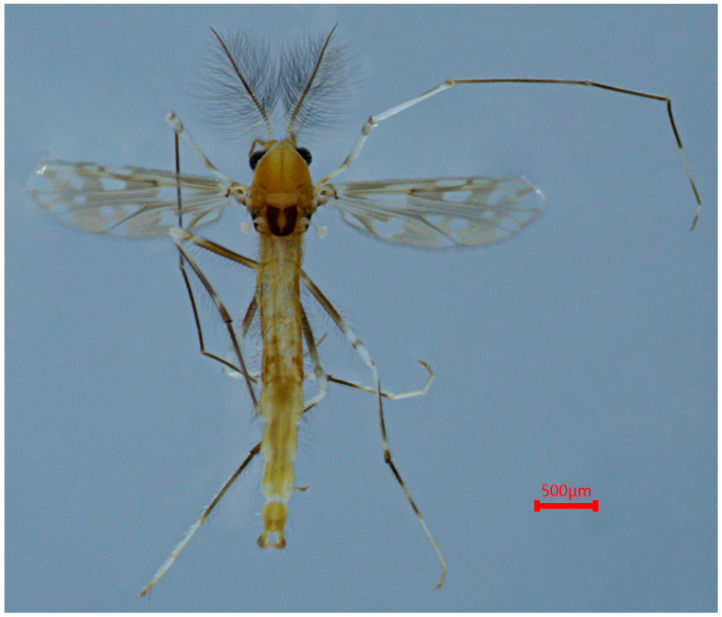
Male adult (holotype, in dorsal view) of *Stictochironomus trifuscipes* sp. nov.

**Figure 7 insects-15-00179-f007:**
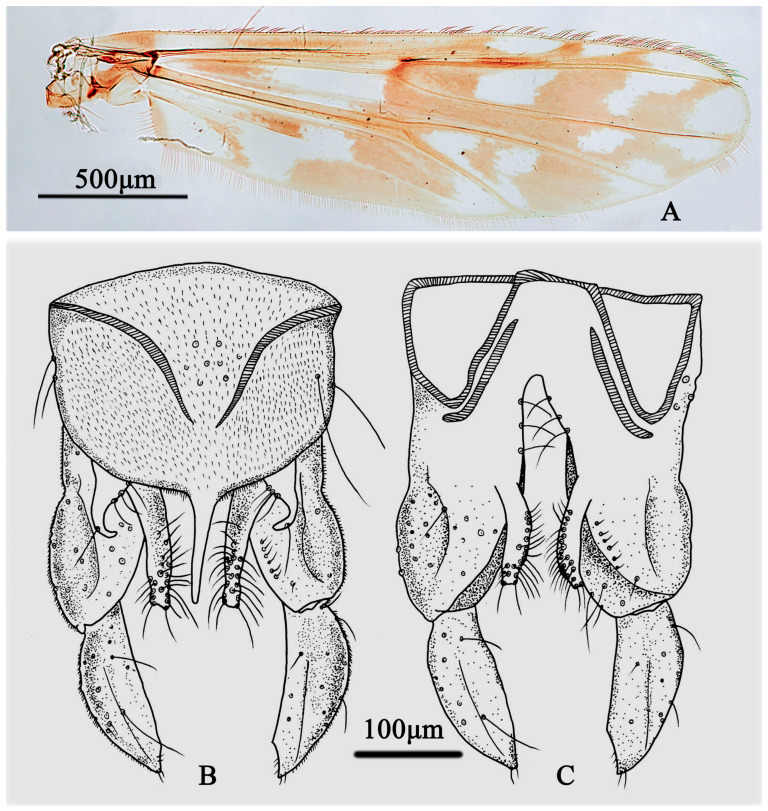
Male adult of *Stictochironomus trifuscipes* sp. nov.: (**A**) wing; (**B**) hypopygium in dorsal view; (**C**) hypopygium in ventral view.

**Table 1 insects-15-00179-t001:** Variable and informative sites, and average nucleotide composition in the aligned COI gene sequences.

Nucleotide Position	Conserved Sites (%)	Variable Sites (%)	Informative Sites (%)	Adenine (%)	Thymine (%)	Cytosine (%)	Guanine (%)
1	42.83	18.20	15.91	28.65	26.23	18.07	27.05
2	52.35	3.00	0.49	13.14	43.57	26.76	16.52
3	4.82	78.80	83.60	39.82	42.36	13.79	4.57
Total	61.70	38.30	33.90	27.02	37.39	19.54	16.05

**Table 2 insects-15-00179-t002:** The highest intraspecific divergence in *Stictochironomus* species; * means only one sample.

No.	Species	Distance (%)	No.	Species	Distance (%)
1	*Stictochironomus akizukii*	15.56	9	*Stictochironomus sinsauensis*	0.77
2	*Stictochironomus devinctus*	2.42	10	*Stictochironomus* sp. 2TE	2.03
3	*Stictochironomus juncai*	14.4	11	*Stictochironomus* sp. INVSJ	0.21
4	*Stictochironomus maculipennis*	1.24	12	*Stictochironomus* sp. T08	4.78
5	*Stictochironomus pictulus*	14.16	13	*Stictochironomus* sp. T09	*
6	*Stictochironomus psilopterus*	3.16	14	*Stictochironomus sticticus*	16.81
7	*Stictochironomus quadrimaculatus*	*	15	*Stictochironomus trifuscipes*	10.85
8	*Stictochironomus rosenschoeldi*	2.68	16	*Stictochironomus unguiculatus*	2.02

**Table 3 insects-15-00179-t003:** Male adults of *Stictochironomus quadrimaculatus* sp. nov. Length (in μm) and proportions of leg (*n* = 1).

	Fe	Ti	Ta_1_	Ta_2_	Ta_3_
P1	1050	670	1260	825	600
P2	1090	870	550	255	200
P3	1100	950	755	370	290
	Ta_4_	Ta_5_	LR	BV	SV
P1	515	255	1.88	1.36	1.37
P2	125	75	0.63	3.83	3.56
P3	200	90	0.79	2.95	2.72

**Table 4 insects-15-00179-t004:** Comparison of main characteristics that show differences between male *S. han* and *S. quadrimaculatus* sp. nov.

Species	Wing Length (mm)	Spots Numbers and Location in Cell				
		r_4+5_	m_1+2_	m_3+4_	cu	an
*S. ha*	3.04	2 downside	0	1 downside	1 downside	1 downside
*S. quadrimaculatus* sp. nov	1.8	2 in the middle	2 in the middle	1 in the middle	1 in the middle	Absent
**Species**	**AR**	**Inferior volsella** **Setae Number**	**P1-LR**	**Leg Pigmentation**		
				**P1-Fe**	**P1-Ti**	**F-Ta_1_**
*S. ha*	2.48	25–26	1.12	1 subapical pale ring	With 2 dark rings	Mostly pale
*S. quadrimaculatus* sp. nov	1.44	12	1.88	3 dark rings	With 4 dark rings	Mostly brown

**Table 5 insects-15-00179-t005:** Male adults of *Stictochironomus trifuscipes* sp. nov. Length (in μm) and proportions of leg (*n* = 2).

	Fe	Ti	ta_1_	ta_2_	ta_3_
P1	1240–1300	940–950	1330–1410	820–850	620–640
P2	1290–1300	950–1000	600–650	290–325	205–235
P3	1310–1410	1000–1090	850–950	430–460	325–345
	ta_4_	ta_5_	LR	BV	SV
P1	520–590	255–260	1.41–1.48	1.56–1.59	1.55–1.68
P2	130–140	125–130	0.63–0.65	3.76–3.97	3.53–3.73
P3	205–210	200–210	0.85–0.87	2.92–2.95	2.62–2.72

**Table 6 insects-15-00179-t006:** Comparison of main characteristics that show differences between male *S. multannulatus*, *S. simantomaculatus* and *S. trifuscipes* sp. nov.

Species	Color of Fore-Leg			Ac	Dc	Scts	Svo Setae	Ivo Setae
	Femur	Tibia	TaI-III					
*S. multannulatus*	most black, 2 pale rings	most dark, 2 pale rings	mostly pale	14–20	10–15	24–28	3 inner, 1 lateral	14
*S. simantomaculatus*	most pale, 3 dark rings	most pale, 2 dark rings	absent	18–20	24–24	28–34	2 inner, 0 lateral	16
*S. trifuscipes* sp. nov	most pale, dark rings	most pale, 3 dark rings	dark brown	8–12	17–20	10–10	3 inner, 1 lateral	18–20

## Data Availability

The molecular data presented in this study are openly available in BOLD (DOI: dx.doi.org/10.5883/DS-STICTO).

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
