# Peer review of "DNA Barcoding Supports “Color-Pattern’’-Based Species of Stictochironomus from China (Diptera: Chironomidae)"

_insects, 2024, doi:10.3390/insects15030179_

Round 1
Reviewer 1 Report (Previous Reviewer 2)
Comments and Suggestions for Authors
The presented article still has serious methodological flaws, especially phylogenetic analysis and species delimitation analysis.
Please specify the protocol for sequencing if it was done by authors or specify if it was ordered by an external company.
Specify why the GTR+G+I model was used in the ML analysis. Was this a result of a test- if so specify the program used for that. Specify the model used in BI analyses.
Why in BI analysis 5 mln generations were used and in Beast 10 mln?
What were the topologies in phylogenetic trees in NJ, ML, and BI- were they the same in all three methods? If so, specify that, if not present all trees.
Specify what "*" is used for in the phylogenetic tree.
If support values/ posterior probability values on major branches in the phylogenetic trees are below estimated values (authors set it on 75/70/0.95 ) presented tree is unreliable- only values on the end of the branches are above that set level conclusion made by authors about "monophyletic cluster" is unsupported in presented data.
Line 175-176 "... As the vouchers were not accessible in our analysis, we conclude that great cryptic species commonly exist in these taxa" All analyzed data should be confirmed and unified. Any assumption cannot be made on unverified data, data without verification should be excluded from the analysis.
The results and discussion sections lack the description of results from phylogenetic analysis. Please complete that.
The results of species delimitation analysis should be summarized on one graph together with the phylogenetic tree which could be easily compared e.g Kajtoch et al. 2018 Mol Phylogenet Evol. 2018 Mar:120:354-363.
The barcode gap threshold should be estimated separately within species and between species, those numbers cannot be presented as the same thing as in the present version of the article.
There are no conclusions after performing a delamination analysis. In the methods section ASAP analysis is described and in the Results and discussion, the ABDG is mentioned. Specify which method ASAP or ABDG was performed.
Does any of the obtained results from GMYC, ABGD, or PTP suggest that S.quadrimaculatus and S. trifuscipens may be considered separate species? Please specify that in the text.
The manuscript in my opinion should be verified by a native speaker. A simple summary and abstract should be rewritten.
Author Response
Dear Reviewer,
Thank you for your helpful comments and sugguestions. Answers are as follows:
Q: Please specify the protocol for sequencing if it was done by authors or specify if it was ordered by an external company.
A: Sequencing was done buy external company, as stated that “and sequenced at Beijing Genomics Institute Co., Ltd., Hangzhou, China.”
Q: Specify why the GTR+G+I model was used in the ML analysis. Was this a result of a test- if so specify the program used for that. Specify the model used in BI analyses.
A: The model “GTR+G+I” was tested in software PartitonFinder, and in BI, model was follows: SYM+G for the first site, F81+I for the second site, and GTR+I+G for the third site.
Q: Why in BI analysis 5 mln generations were used and in Beast 10 mln?
A: In BI analysis, we terminated when the standard deviation of split frequencies is below 0.01, when 5 million generations, while In beast, we check the trace file, when the ESS > 200 of all parameters, we stop the running.
Q: What were the topologies in phylogenetic trees in NJ, ML, and BI- were they the same in all three methods? If so, specify that, if not present all trees.
A: All the three trees are almost sharing the same topology, and we provide all the trees in the supplement files.
Q: Specify what "*" is used for in the phylogenetic tree.
A: * means that no support indicated on nodes.
Q: If support values/ posterior probability values on major branches in the phylogenetic trees are below estimated values (authors set it on 75/70/0.95) presented tree is unreliable- only values on the end of the branches are above that set level conclusion made by authors about "monophyletic cluster" is unsupported in presented data.
A: It is true that the stem branch is low, and end branches with high support values for the high variation rate of COI. The branch cluster as we can see in Figure 1 is species level, which indicate that different species can be divided into distinct clades. The question may the relationships between different species could not be inferred, but this is not the main points of our ms.
Q: Line 175-176 "... As the vouchers were not accessible in our analysis, we conclude that great cryptic species commonly exist in these taxa” All analyzed data should be confirmed and unified. Any assumption cannot be made on unverified data, data without verification should be excluded from the analysis.
A: We delete such description.
Q: The results and discussion sections lack the description of results from phylogenetic analysis. Please complete that.
A: We have completed “Phylogenetic trees of neighbor-joining (S5), maximum likelihood (S6), and Bayesian inference are almost shared the same topology (S7). Relationships between species are not well constructed as high interspecific variation in COI (Ekrem et al. 2010). However, the large interspecific variation aids the diagnosis of species clusters, therefore high support values on the end of the branches. Based on the NJ tree, ML tree, and BI tree, 218 DNA barcodes of 16 initial morphospecies were clustered into 22 clades (Figure 1). The undescribed taxa here well form separated clades in all phylogenic trees, that support them as new to science.”
Q: The results of species delimitation analysis should be summarized on one graph together with the phylogenetic tree which could be easily compared e.g Kajtoch et al. 2018 Mol Phylogenet Evol. 2018 Mar:120:354-363.
A: we have summarized in one para.
Q: The barcode gap threshold should be estimated separately within species and between species, those numbers cannot be presented as the same thing as in the present version of the article.
A: In the ASAP analysis, we detect the gap, and from the gap a distance threshold is estimate and used to partition the samples into putative species, and seen in Mol Ecol Resour. 2021;21:609–620. ASAP: assemble species by automatic partitioning,
Q: There are no conclusions after performing a delamination analysis.
A: The conclusion is “Species delimitation analysis using ASAP, PTP, GMYC well support the two new undescribed species to science”that added in conclusion part.
Q: In the methods section ASAP analysis is described and, in the Results, and discussion, the ABDG is mentioned. Specify which method ASAP or ABDG was performed.
A: It should be ASAP, and we made it clear in the ms.
Q: Does any of the obtained results from GMYC, ABGD, or PTP suggest that S.quadrimaculatus and S. trifuscipens may be considered separate species? Please specify that in the text.
A: Yes, both GMYC, PTP, and asap suggest that S.quadrimaculatus and S. trifuscipens should be considered separate species. We describe that in the ms.
Q: The manuscript in my opinion should be verified by a native speaker. A simple summary and abstract should be rewritten.
A: We have had our ms reviewed by Jack David Bamber, the English editor of MDPI, and abstract and summary are changed.
Thank you in advance.
Cheers
Chao
Reviewer 2 Report (Previous Reviewer 3)
Comments and Suggestions for Authors
The manuscript “DNA barcoding supports “color-pattern’’ -based species of Stictochironomus from China (Diptera: Chironomidae)” provides a morphological and genetic analysis of the genus and describes 2 new species. This research discusses the use of color patterns to distinguish species in Stictochironomus and recommends a different threshold to delimit species in the genus. Overall, the research is a valuable contribution that adds to our understanding of the taxonomy of Stictochironomus. The revisions the author’s have undertaken greatly improve the manuscript and address the concerns I had with a previous version of this manuscript. The study design is sufficiently described in the manuscript. My detailed comments and suggestions are provided below. I recommend that this manuscript be accepted with minor revisions.
Comments on the Quality of English Language
The grammar in this manuscript requires some revising, but overall, the grammar is good. My comments on this draft are largely related to improving the grammar. They are as follows.
Page 4, line 146: delete “obviously”, not needed
Page 4, lines 147-150: I suggest rewording this sentence – “The high intraspecific divergence in our study can be attributed to four species (Stictochironomus sticticus (Fabricius): 16.81%; Stictochironomus pictulus (Meigen): 14.41%; Stictochironomus akizuki (Tokunaga) 15.56%; and Stictochironomus juncaii Qin, Shi & Wang: 9.35%; Table 2).”
Page 5, line 154: There should be a space between “species” and “have.” There are missing spaces throughout the manuscript and this should be corrected.
Page 5, line 158: I suggest rewording this sentence – “These vouchers should be rechecked to determine if incorrect identifications are responsible for the high intraspecific divergence we observed for some Stictochironomus species.”
Page 5, lines 163-165: The meaning of this sentence is not clear. I have suggested a rewording for this sentence, but the authors should ensure that this is consistent with their intent. “In our study, we did not observe interspecific overlap of color pattern morphotypes, which could support color patterns as a good diagnostic characteristic for species delimitation in Stictochironomus. However, species with unique color morphotypes could form several separated clades based on genetic analyses.”
Page 5, line 166-168: I suggest rewording this sentence – “Four species with unique color patterns formed several phylogenetic clades (Figure 1) which matches the results in the thesis of Reistad (2023) [39]: These four species were the same species with high intraspecific divergence and included S. pictulus (four clades), S. sticticus (three clades), S. akizukii (two clades), and S. juncaii (two clades).”
Page 5, line 171: delete “an”
Page 5, line 176: delete “great”
Page 5, lines 177-179: I suggest rewording this sentence – “Most barcoding research on insect species proposes a threshold 2-3% to delimit intra- or interspecific species [41–43]; however, genus-specific thresholds have been suggested for some Chironomidae including 4–5% for Tanytarsus [35] and 5–8% for Polypedilum [23].”
Page 6, line 203: I suggest deleting “, on the one hand.” It is not necessarily incorrect, but it’s not needed.
Pages 8-9, lines 232-246: This text is repeated.
Page 9, lines 271-278: I suggest rewording this sentence – “The new species is similar to Stictochironomus han Na & Bae, 2010. Both species share the following characteristics: wing with obvious separated spots; anal point long, slender, nearly parallel-sided, with rounded apex; superior volsella broad basally with microtrichiae and three long setae on inner margin and with the long distal process slightly curved, bare, with hooked apex and a long seta on dorsal surface in distal 1/3 [44]. Stictochironomus quadrimaculatus sp. nov. differs from S. han by a lower AR (1.44), shorter wing (1.80mm), larger foreleg LR (1.88), and having unique pigmentation patterns on the legs and wings, such as spot locations, sizes, and numbers (Figure 1A and 1D in [44]). In S. han, these measurements are AR 2.48, wing length 3.04 mm, and forleg LR 1.12.”
Page 12, lines 323-334: This text is repeated
Page 13, lines 355-366: I suggest rewording this sentence. The characters should be checked to ensure that they are accurate following these revisions. “The new species is similar to Stictochironomus simantomaculatus (Sasa, Suzuki & Sakai, 1998). Both species have similar wing patterns and pigmentation of the fore femur and fore tibia, but they can be differentiated by the morphology of the hypopygium. Specifically, the superior volsella has three inner setae and one lateral outer seta in S. trifuscipes sp. nov. versus two inner setae and zero lateral outer seta in S. simantomaculatus. In addition, the gonostylus is normally developed in S. trifuscipes sp. nov. and abruptly constricted at about the middle in S. simantomaculatus. The wing patterns and superior volsellae of Stictochironomus trifuscipes sp. nov. also resemble Stictochironomus multannulatus (Tokunaga, 1938) (Figure 38–39 in [45]), but these species can be distinguished by leg pigmentation and details on the inferior volsella. For example, ta1–3 of P1 is totally dark brown (Figure 5B) while it is mostly pale in the latter species. In addition, the inferior volsella has 18–20 recurved setae versus 16 setae, the gonostylus is without long strong setae along the inner margin versus with strong setae on the ventral surface and several long bristles arising from the basal half (Table 6). In addition, the average species k2p distance to the other Stictochironomus ranged from 16% to 24% (S4), which to some extent, supports it as a new species.”
Page 16, lines 478-482: I suggest rewording this sentence – “Since it can be prohibitive to check the vouchers registered in BOLD or GenBank, we urge scientists to confirm their identifications with experts and to upload supplementary information, such as closeup images of voucher specimens and habitat descriptions.”
Author Response
Dear reviewer,
Thank you for your helpful comments and suggustions.
We agree with the suggestions that you made in the ms, and we accepted that changes.
Thank you so much.
Chao
Round 2
Reviewer 1 Report (Previous Reviewer 2)
Comments and Suggestions for Authors
The article in its present form meets all my requirements and can be published.
This manuscript is a resubmission of an earlier submission. The following is a list of the peer review reports and author responses from that submission.
Round 1
Reviewer 1 Report
Comments and Suggestions for Authors
1. On page 10 in the Distribution section add “Russian Far East (Orel, 2016)" and in References add: Orel, O.V. (2016) Fauna of non-biting midges of subfamily Chironominae (Diptera, Chironomidae) of the Russian Far East. Freshwater Life, 2, 185–196. [In Russian]
2. In Abstract change the sentence “S. trifuscipes Song & Qi, sp. nov. and S. quadrimaculatus Song & Qi, sp. nov.” to S. trifuscipes sp. nov. and S. quadrimaculatus sp. nov.
3. On page 5 change the sentence "3.2.1. Stictochironomus quadrimaculatus Song & Qi, sp. nov." to 3.2.1. Stictochironomus quadrimaculatus Song et Qi, sp. nov. Also, you need to register a new species in ZooBank.
4. On page 7 the same - "3.2.2. Stictochironomus trifuscipes Song & Qi, sp. nov." to 3.2.2. Stictochironomus trifuscipes Song et Qi, sp. nov. And registration in ZooBank.
5. On page 3, lines 121-122: sentence "These vouchers should await rechecking to make a final 121 decision." written 2 times.
Author Response
Dear,
Questons and answers were below.
Question. Comments:1. On page 10 in the Distribution section add “Russian Far East (Orel, 2016)" and in References add: Orel, O.V. (2016) Fauna of non-biting midges of subfamily Chironominae (Diptera, Chironomidae) of the Russian Far East. Freshwater Life, 2, 185–196. [In Russian]
Answer. The literature has been added to the references.
Question. Comment: 2. In Abstract change the sentence “S. trifuscipes Song & Qi, sp. nov. and S. quadrimaculatus Song & Qi, sp. nov.” to S. trifuscipes sp. nov. and S. quadrimaculatus sp. nov..
Answer: we have accepted the suggestion.
Question. Comment: 3. On page 5 change the sentence "3.2.1. Stictochironomus quadrimaculatus Song & Qi, sp. nov." to 3.2.1. Stictochironomus quadrimaculatus Song et Qi, sp. nov. Also, you need to register a new species in ZooBank.
On page 7 the same - "3.2.2. Stictochironomus trifuscipes Song & Qi, sp. nov." to 3.2.2. Stictochironomus trifuscipes Song et Qi, sp. nov. And registration in ZooBank.
Answer. We have changed the sentence, and ZooBank register accessions were added.
Stictochironomus quadrimaculatus Song et Qi, sp. nov.
urn:lsid:zoobank.org:act:EC674AA4-48E7-4D2E-A869-B00FE884EA98
Stictochironomus trifuscipes Song et Qi, sp. nov.
urn:lsid:zoobank.org:pub:7B3EC4A2-83A9-4618-91DA-75C8022EAB96
Question. Comment: 4. . On page 3, lines 121-122: sentence "These vouchers should await rechecking to make a final 121 decision." written 2 times.
Answer. We have deleted duplicates.
Reviewer 2 Report
Comments and Suggestions for Authors
Song et al. reported that DNA barcodes support "color pattern-based species in Strictochironomus from China. Hovewer phylogenetic analysis if full of faults. First, the phylogenetic tree should be rooted in the outgroup, without that you are not getting variable phylogenetic, and yours conclusions are not accurate. Whats more phylogenetic analyses should be performed using more reliable methods, such as Maximum Likelihood or Bayesian Interference, the best would be both of them. Phylogenetic analysis was done sloopy. The authors mentioned that genetic distances were calculated, but they are not added to the supplement, and the values for new species are not mentioned in the results.
Phylogenetic analysis was performed on 218 sequences, but the authors mentioned 702 sequences twice, which corresponds to the number of sequences downloaded from BOLD. The most important factor was the number of sequences used for analysis.
For large alignments, it is best to use other alignment metonds that ClustaW, such as MAFFT or even MUSCLE.
In the Methods section it should be specified from how mamy samples were DNA isolated, authors are mentionig that it was isolated from heads and thorax- for all samples? Methods section lacks information on the method used for DNA purification and a description of the sequencing conditions and protocol.
Author Response
Dear,
Here are the answers.
Question. Comment:1. First, the phylogenetic tree should be rooted in the outgroup, without that you are not getting variable phylogenetic, and your conclusions are not accurate.
Answer. Generally, barcoding studies pay more attention on species delimitation less on phylogeny relationships. Outgroup rooting is not a necessary, but yes outgroup rooting could be done, and we use Stenochironomus species as outgroup (updated tree of figure 1).
Question. Comment:2. What’s more phylogenetic analyses should be performed using more reliable methods, such as Maximum Likelihood or Bayesian Interference, the best would be both of them.
Answer. ML and BI inferences were done and updated in Figure 1. Species delimitation analysis used ASAP, PTP, GMYC to conclude more reliable results.
Question. The authors mentioned that genetic distances were calculated, but they are not added to the supplement, and the values for new species are not mentioned in the results.
Answer. Genetic distances were added to supplement as S4: Average interspecific divergences (k2p distances). Also mentioned in Remarks.
Question. Phylogenetic analysis was performed on 218 sequences, but the authors mentioned 702 sequences twice, which corresponds to the number of sequences downloaded from BOLD. The most important factor was the number of sequences used for analysis.
Answer. We made it clear that Three datasets were compiled for the analysis: dataset (A) raw data (704 sequences, S1); dataset (B) without duplicated sequence (218 sequences, S2); dataset (C) holotype sequences (75 sequences, S3). Datasets A and C were only used for ASAP analysis.
Question. For large alignments, it is best to use other alignment methods that ClustaW, such as MAFFT or even MUSCLE.
Answer. We used ClustalW in mega 7.
Question. In the Methods section it should be specified from how many samples were DNA isolated, authors are mentioning that it was isolated from heads and thorax- for all samples?
Answer. In addition to our own data (GenBank accession numbers: OR822593-OR822597), available Stictochironomus COI barcodes in BOLD were searched, and 704 sequences (in all) were added to the dataset named “Barcodes of Stictochironomus species. Our data (OR822593-OR822597) are isolated from heads and thorax. As to the others are unknown.
Question. Methods section lacks information on the method used for DNA purification and a description of the sequencing conditions and protocol.
Answer: DNA amplification was carried out in a 25 µl volume, including 12.5 µl 2×Es Taq MasterMix (CoWin Biotech Co., Beijing, China), 0.625 µl of each primer, 2 µl of template DNA and 9.25 µl deionized H2O. PCR amplification was an initial denaturation step of 95°C for 5 min, followed by 34 cycles of 94°C for 30 s, 51°C for 30 s, 72°C for 1 min, and final extension at 72°C for 3 min following Song et al. (2018) [23]. PCR products were electrophoresed in 1.0% agarose gel, purified with ExoProStar 1-Step, and sequenced using an ABI 3730XL capillary sequencer (Beijing Genomics Institute Co., Ltd., Hangzhou, China).
But sequencing was done by the Beijing Genomics Institute Co., Ltd., Hangzhou, China, and detailed protocol was not provided by the company.
Cheers
Reviewer 3 Report
Comments and Suggestions for Authors
The manuscript “DNA barcoding support “color-pattern” based species on Stictochironomus from China (Diptera: Chironomidae)” provides a phylogenic tree based on available COI barcodes and describes 2 new species for the genus. Overall, the research is a potentially valuable contribution that could add to our understanding of the taxonomy of Stictochironomus. The study design is sufficiently described in the manuscript. The grammar in this manuscript requires revising. My detailed comments and suggestions are provided below. I recommend that this manuscript be rejected. It may be suitable for resubmission with major revisions.
Specific comments:
My largest concern is that the study proports to identify a congruence between “color-pattern” and barcoding results. However, this is not broadly demonstrated within the genus by this research, and I have concerns regarding the erection of two species largely based on color pattern alone. The authors’ state that color patterns in Stictochironomus are plastic which then requires demonstration that unique and consistent patterns can be identified to separate species or clades. Overall, I don’t see this demonstrated by this research. The authors’ objectives are perhaps best attained with the description of S. quadrimaculatus sp. nov. because there is good genetic differentiation of this species. However, the description of the species relies on a single specimen and is morphologically based on the color pattern alone. The second described species, S. trifuscipes sp. nov., is based solely on color pattern. The similar S. multannulatus was not compared genetically and it is only separated from S. trifuscipes sp. nov. using the leg pattern. Given the variability of patterning in Stictochironomus, it seems that relying on this to erect a species is tenuous. It would be useful if a more detailed analysis of the morphology were performed. Cryptic species that are recognized genetically can often then be examined in more detail to identify morphological characters or a combination of characters which separate the species.
Page 1, line 27: delete “frequent”
Page 2, lines 70-71: Suggest rewording - “there are about more than 40” to “there are more than 30”
Page 2, line 90: change “followed” to “following”
Page 3, line 112: delete “the” before Stictochironomus
Page 3, lines 119-139: This section needs to be reworded to improve the grammar.
Page 3: Since this research indicates that color pattern is a useful character for separating Stictochironomus species, a more in-depth analysis of this argument may be helpful. For example, is it possible to compile pattern information for species and discuss how this relates to the phylogenetic tree obtained in this study? In its current form, it is not clear how useful color patterns are for separating species in this genus. Perhaps a table documenting the color patterns and variability of these patterns for the different species and clades would be helpful? However, this may be difficult given the lack of supplemental information provided for some specimens in existing databases.
Pages 6 and 8, Figures 3 and 5: The illustrations of the hypopygium are small. It would be helpful if these illustrations were larger.
Page 10, lines 293-294: Suggest rewording - “Regarding the superior volsella wing, there is one basal seta” to “The superior volsella has one basal seta”
Page 11, line 365: This couplet references Figure A15 in [31]. The reference [31] does not have a figure of Stictochironomus pictulus. Correct the reference.
Page 11, line 365: Stictochironomus pictulus is spelled incorrectly.
Page 11: Color patterns are heavily relied upon in the provided key to Stictochironomus species known from China. These color patterns are presumably based on the original descriptions and additional material examined by the authors. Are they variable within or between species? Could other morphological characters be included in the key?
Page 12: It may be useful to review the following as well: Reistad, I. (2023). A Review of Fennoscandian Stictochironomus (Diptera: Chironomidae), with the description of a species new to science. This thesis discusses some of the variability in S. pictulus and S. sticticus.
Comments on the Quality of English LanguageThe grammar in this manuscript requires moderate revising. Throughout the manuscript, there are minor to moderate edits needed. Specifically, lines 119-139 need to be reworded to improve the grammar. I have provided some suggestions to improve the grammar, but these comments are not exhaustive.
Author Response
Dear,
Here are the answers.
Question. The grammar in this manuscript requires revising.
Answer. Our ms has undergone English language editing by MDPI.
Question. My largest concern is that the study proports to identify a congruence between “color-pattern” and barcoding results.
Answer. In our phylogeny trees, the two new described species with unique color patterns form distinctly separate clade. The species form several separate clades, which may indicate potential cryptic diversity in the future research.
Question.I have concerns regarding the erection of two species largely based on color pattern alone. The authors’ state that color patterns in Stictochironomus are plastic which then requires demonstration that unique and consistent patterns can be identified to separate species or clades. Overall, I don’t see this demonstrated by this research.
Answer. We have provided other characteristics to help delimitate species as in Table 4 and Table 6, such as Ac, Dc, Scts, superior volsella, and inferior volsella. In the key, we could see the with spot or without or even the spots numbers of the wing is very important, in the supplement file S10, it clearly showed the each species with unique color patterns.
Question. Given the variability of patterning in Stictochironomus, it seems that relying on this to erect a species is tenuous.
Answer. We provided Ac, Dc, Scts, superior volsella, and inferior volsella to delimit them.
Question. Page 1, line 27: delete “frequent”;Page 2, lines 70-71: Suggest rewording - “there are about more than 40” to “there are more than 30”;Page 2, line 90: change “followed” to “following”;Page 3, line 112: delete “the” before Stictochironomus
Answer. We accepted the suggestions.
Question. Page 3: Since this research indicates that color pattern is a useful character for separating Stictochironomus species, a more in-depth analysis of this argument may be helpful. For example, is it possible to compile pattern information for species and discuss how this relates to the phylogenetic tree obtained in this study? In its current form, it is not clear how useful color patterns are for separating species in this genus. Perhaps a table documenting the color patterns and variability of these patterns for the different species and clades would be helpful? However, this may be difficult given the lack of supplemental information provided for some specimens in existing databases.
Answer. Yes, we provide color pattern table in S10 of Stictochironomus from China, from the table we could clearly see that each species with different color pattern.
Question. Page 3, lines 119-139: This section needs to be reworded to improve the grammar.
Answer. We have reviewed the para.
Question. Pages 6 and 8, Figures 3 and 5: The illustrations of the hypopygium are small. It would be helpful if these illustrations were larger.
Answer. We have made larger illustrations as in updated figure.
Question. Page 11: Color patterns are heavily relied upon in the provided key to Stictochironomus species known from China. These color patterns are presumably based on the original descriptions and additional material examined by the authors. Are they variable within or between species? Could other morphological characters be included in the key?
Answer. Color patterns are variable between species. And we have provide other morphological characters in the updated key.
Question. Page 12: It may be useful to review the following as well: Reistad, I. (2023). A Review of Fennoscandian Stictochironomus (Diptera: Chironomidae), with the description of a species new to science. This thesis discusses some of the variability in S. pictulus and S. sticticus
Answer. We have reviewed the thesis of Reistad, I. (2023), and found the setae numbers on thorax are important in species delimitation.
Question. This couplet references Figure A15 in [31]. The reference [31] does not have a figure of Stictochironomus pictulus. Correct the reference.
Answer. We have changed the references: Sasa, M; Suzuk, H. Studies on Chironomid midges collected in Hokkaido and Northen Honshu. Trop. Med. 1998, 40, 9-43.
Cheers